

# Current-induced magnetization hysteresis defines atom trapping in a superconducting atomchip

**Fritz Diorico[1*], Stefan Minniberger, Thomas Weigner, Benedikt Gerstenecker, Naz Shokrani, Żaneta Kurpias and Jörg Schmiedmayer[†]**

Vienna Center for Quantum Science and Technology, Atominstitut TU Wien, Stadionallee 2, 1020 Vienna, Austria

⋆ fritz.diorico@tuwien.ac.at  † schmiedmayer@atomchip.org

## Abstract

The physics of superconducting films, and especially the role of remanent magnetization has a defining influence on the magnetic fields used to hold and manipulate atoms on superconducting atomchips. We magnetically trap ultracold $^{87}$Rb atoms on a 200 μm wide and 500 nm thick cryogenically cooled niobium Z-wire structure. By measuring the distance of the atomcloud to the trapping wire for different transport currents and bias fields, we probe the trapping characteristics of the niobium superconducting structure. At distances closer than the trapping wire width, we observe a different behaviour than that of normal conducting wire traps. Furthermore, we measure a stable magnetic trap at zero transport current. These observations point to the presence of a remanent magnetization in our niobium film which is induced by a transport current. This current-induced magnetization defines the trap close to the chip surface. Our measurements agree very well with an analytic prediction based on the critical state model (CSM). Our results provide a new tool to control atom trapping on superconducting atomchips by designing the current distribution through its current history.


# 1 Introduction

Trapping and manipulating neutral atoms on atomchips [1–3] plays an important role in probing fundamental quantum science and developing measurement tools. Its portable nature makes it ideal for metrology and quantum technology applications. The majority of the existing atomchip experiments are structured from normal conducting materials. In recent years the first atomchips operating with superconducting wires were implemented [4–27].

Superconducting atomchips are a promising tool for many aspects in atom optics and atom based quantum technology. They provide advantages like reduced Johnson-Nyquist noise [9, 25, 28] compared to normal conducting wires [29] and the possibility to create a noise-free trap without any external connection to provide a transport current [6, 21, 23, 26]. Bringing ultracold atoms close to superconducting resonators allows to create hybrid quantum systems [17, 30, 31], coupling superconducting quantum electronics [32] to atomic quantum memory. They can also form a platform to study Rydberg atoms [10, 33, 34] and create novel superconducting trap arrays for quantum simulation [35, 36]. Furthermore, superconducting atomchips were used to probe properties such as lifetime enhancement (reduced Johnson-Nyquist noise), magnetization hysteresis, flux quanta and the Meissner effect [8, 9, 14–16].

For all these envisioned developments and applications, it is of utmost importance to understand the role of the intricate physics that superconductivity plays in creating the traps used to store and manipulate the atoms. This is especially true for the complex behaviour of type-II superconducting films with non-trivial geometries. One key ingredient thereby is the remanent magnetization. Previous studies have already investigated the dependence of superconducting atomchip traps on the history of externally applied magnetic fields [7, 11, 14, 24].

In this report, we probe the specifics of the atom traps created by the currents in our superconducting wire by measuring the position of the ultracold atoms relative to the wire and the chip surface. We show that a current-induced remanent magnetization plays an important role in defining the atom trapping and manipulation on a $200\,\mu$m wide superconducting niobium wire. This current-induced remanent magnetization is an effect of the type-II superconductivity of the niobium structure. It will add new tools to tailoring current distributions in superconducting magnetic traps.

# 2 Experimental setup

Our cryogenic atomchip setup is described in great detail in [37, 38]. It consists of a standard magneto-optical trap (MOT) to collect and cool $^{87}$Rb atoms and a magnetic conveyor belt to transport the ultracold $^{87}$Rb atoms into a cryogenic environment, where they are loaded into the superconducting atomchip trap (See figure 1). The magnetic transport scheme to bring ultracold atoms into the cryostat comprises of a series of current pulses on successive coil pairs.

The superconducting atomchip used in the experiments reported here contains a simple niobium Z structure with a width of $200\,\mu$m and a length of $2\,$mm (See figure 1b). The $500\,$nm thick niobium film was deposited by sputtering onto a sapphire substrate. We measure a critical temperature of $9.1\,$K. The atoms trapped below this structure are detected by in-situ absorption imaging to resolve the trapped atom density distribution. The imaging beam is incident at an oblique angle with respect to the chip surface to produce a reflection image. This reflection image allows measurement of the distance of the atomcloud to the atomchip surface irregardless of an off-axis (z-axis) tilt of the atoms towards trapping wire. See [39] for more details on the imaging technique.

The atomchip trap is formed by adding a homogeneous, horizontal magnetic bias field $B_{bias}$, parallel to the chip surface and perpendicular to the current $I_t$ in the central region of

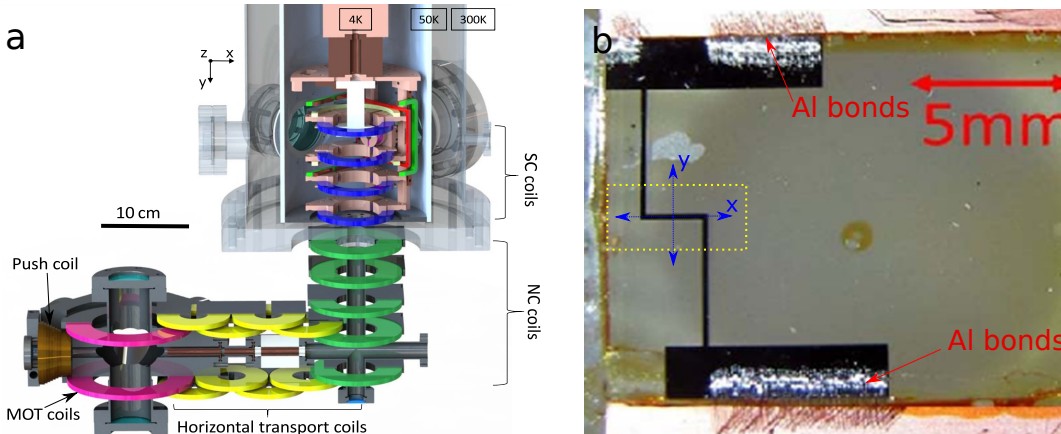

Figure 1: (a), sketch of the experimental setup adapted from [37]. A magnetic conveyor belt transport which brings ultracold atoms to a cryogenic environment. Its components are: Push coil (brown), MOT coils (pink), horizontal transport coils (yellow), normal conducting vertical transport coils (green) and superconducting transport coils (blue). (b), a niobium atomchip on a sapphire substrate. The central region of the Z-wire, roughly defined by the yellow dashed box, indicates the region where the remanent magnetization remains even after exceeding $I_c$, when measuring the critical current in our setup. See section 4.2.

the Z-wire of the chip [1]. The combined magnetic field has a local minimum at a distance $d$ from the chip surface where the bias field cancels (up to a longitudinal component orthogonal to the bias field) with the field created by the current in the wire. For an infinitely thin wire, this distance $d$ is given by $d = \mu_o I_t/(2\pi B_{bias})$. For a wire with finite width $2w$[1] the distance $d$ depends on the details of the current distribution in the wire. For the rest of our manuscript we define the distance normalized to the wire width as $d_{2w} = d/2w$. For $d_{2w} \gg 1$, the thin wire approximation describes the behavior of the trap sufficiently, where for $d_{2w} < 1$ the details of the current distribution in the wire become important. We probe distances down to $40\,\mu$m for a $2w = 200\,\mu$m wire ($d_{2w} \sim 0.2$).

## 3 Basic observation

Figure 2 shows measurements of the distances of the trapped atom cloud from the wire versus the applied bias field of up to 70 G (7.0 mT) for a transport current of 1.9 A. The figure also includes measurements for a stable trap at zero transport current. The details of this zero-current trap are also discussed at the end of this section and the following section.

We first compare the experimental data for 1.9 A to a Meissner-London current distribution in the wire [40, 41]. In this case, the superconductor is assumed to be in the Meissner state and the current tends to flow in the edges of the wire to minimize the magnetic field within the superconductor. For a wide wire cross-section the central region has a low current density which makes the trap come close to the wire faster for an increasing bias field.[2] In addition to this the trap opens up toward the chip surface if it gets closer than $d_{2w} \approx 0.2$. In our case this would be for a bias field of 35 G (3.5 mT). This is illustrated by the thick marker at the end of

---

[1]We use $2w$ for the full width of the wire to be consistent with the convention used in [40] (see also inset of figure 2).

[2]The Meissner effect on a cylindrical niobium wire, 125 μm in diameter, was studied in [14] where the Meissner effect was found to shorten the distance of the atoms to the trapping wire.

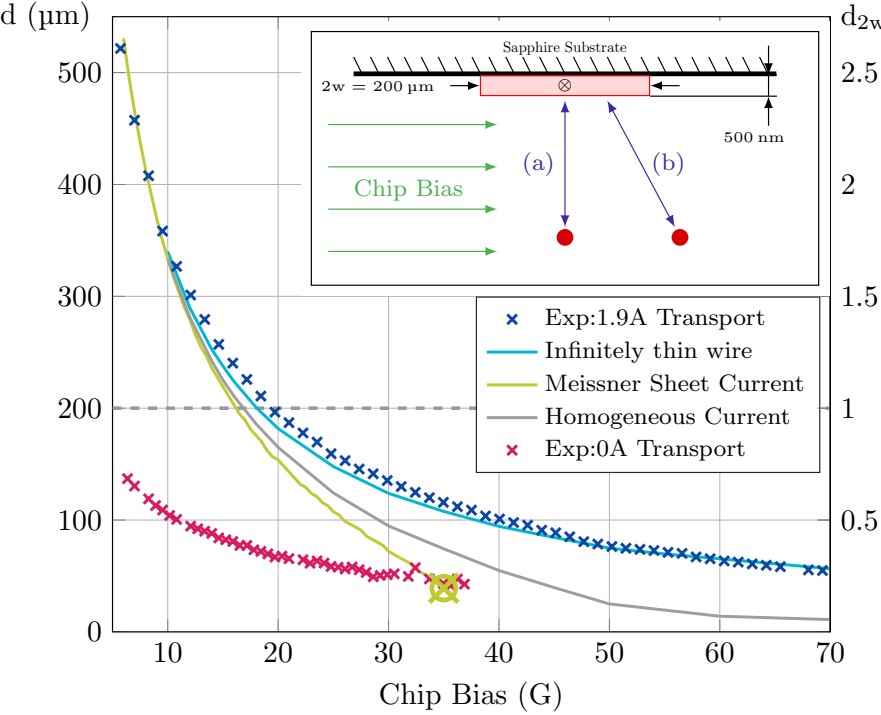

Figure 2: Distance of the atomcloud to the chip surface versus the applied bias field. The discrete crosses and circles are measurements for an applied transport current of 1.9 A and 0 A, respectively. The target trapping current is always ramped from zero. The solid curves shown are the position of the field minimum from the full 3D simulation of the wire geometry according to various cross-sectional current distributions. The yellow-green curve assumes a Meissner-London current distribution [41]. The gray curve is for a homogeneous current across the 200 μm width. The light-blue curve is for a single thin wire. The dashed line illustrates the boundary where $d_{2w} = 1$ ($d = 2w$). The inset shows a basic diagram of the trapping wire with an applied current plus an external bias field. (a) and (b) shows the direction of the trap relative to the wire for a purely current-induced and field-induced magnetization, respectively. See section 4.2 of the text. The measured $d$ always gives the distance of the atoms to the atomchip surface irregardless of its relative position [39].

the yellow-green curve in figure 2. The data shown for 1.9 A is displaying a strictly different behaviour.

The next comparison is with a homogeneous current flow throughout the wire cross-section as in a normal conducting atomchip. Again, due to the wide wire width, the trap distance to the wire surface is expected to decrease faster with increasing bias field than that of a single filament wire trap. Unlike the Meissner current trap discussed above, the trap remains closed for increasing applied bias field up to the fundamental limit (Casimir-Polder forces) [29, 42]. There is still a significant difference between the model (gray line in Fig. 2) and the measurement.

Surprisingly, the model that fits the measurement best is a simple single thin wire. This indicates that transport current in the wire cross-section is flowing more in the center of the wire than at the wire edges, making it behave like a thin wire. This hints at a superconducting effect in our atomchip trap causing an inhomogeneous current across the wire cross-section.

Reducing the transport current from 1.9 A down to 0 A one still reproduces an atomchip trap as shown in figure 2. The observation of a zero-current-trap indicates the presence of a

remanent magnetization induced in the niobium film. As we argue in the following section, the remanent magnetization is induced mainly by the transport currents whereas in previous experiments only field induced remanent magnetization was studied [21].

For atomchip traps created by an applied transport current through a superconducting wire, there is no thorough study that probed the distances smaller than the wire width ($d_{2w} < 1$), where the details of the current distribution plays an important role. In the next section, we will discuss the relevance of the remanent magnetization to atom trapping in this distance range.

## 4  Impact of the remanent magnetization to atom trapping

As discussed earlier, for $d_{2w} < 1$, characteristics of the magnetic trap depends on the specific details of the cross-sectional current distribution. Applying magnetic fields to type-II superconductors induces a remanent magnetization created by permanent super-currents across the wire cross-section [43–45], and, with it, modifies the trapping. Experiments performed with YBCO superconducting atomchips focused on inducing a remanent magnetization with an external magnetic field [21–24]. The study in [8] looked at the remanent magnetization effects of a 200 μm wide niobium film for various magnetic field histories during the experimental cycle.

One of the widely used phenomenological models for the remanent magnetization of type-II superconductors is the Bean critical state model (CSM) [43,44]. The CSM assumes a vanishing first critical field $H_{c1}$ and only treats the thermodynamic critical field $H_c$. Flux penetration is formed through current flow in the edges at the critical current density $J_c$ which, microscopically, is manifested by vortices. If the applied magnetic field goes above $H_c$, a remanent magnetization is formed in the superconductor even after switching off the field. This can be viewed as counter propagating currents with $-J_c$ and $J_c$ at the respective edges of the wire cross-section. This remanent magnetization is analogous to magnetic field hysteresis found in ferromagnets. The current density, $\vec{j}$ and magnetization, $\vec{M}$ obey $\vec{j} = \nabla \times \vec{M}$. Analytical models for the magnetization currents for type-II superconducting rectangular thin films extending to infinity (Brandt model) can be found in [40,46].

The central cross-section of our Z-structure in figure 1 can be treated as a rectangular cross-section extending to infinity. For thin films, the current distribution along the film thickness can be neglected. In this case, only the current distribution across the wire cross-section defines the magnetic trapping of the atoms. This approximation was applied successfully in [8] where the effect of CSM from the wide niobium film beside the trapping wire was observed with ultracold atoms for various magnetic field histories experienced by the entire atomchip. CSM, using the same approximation, was used to study the implications of type-II effects to produce field-induced remanent magnetization magnetic traps with YBCO superconducting atomchips in [21–23,47].

Figure 3 shows a history of the magnetic fields perpendicular to the film experienced by the superconducting atomchip directly at the central section of the Z-wire (See figure 1b) during one experimental cycle. Only fields perpendicular to the film induces a remanent magnetization. Since the niobium film is 500 nm thick, the thin-film approximation applies and the film is transparent to magnetic fields parallel to the film. Varying the perpendicular magnetic field history of the superconducting film induces a different remanent magnetization.

### 4.1  Current-induced magnetization

Inducing a remanent magnetization in a type-II superconductor is usually done with a magnetic field. However, a remanent magnetization can also be formed by an applied transport current.

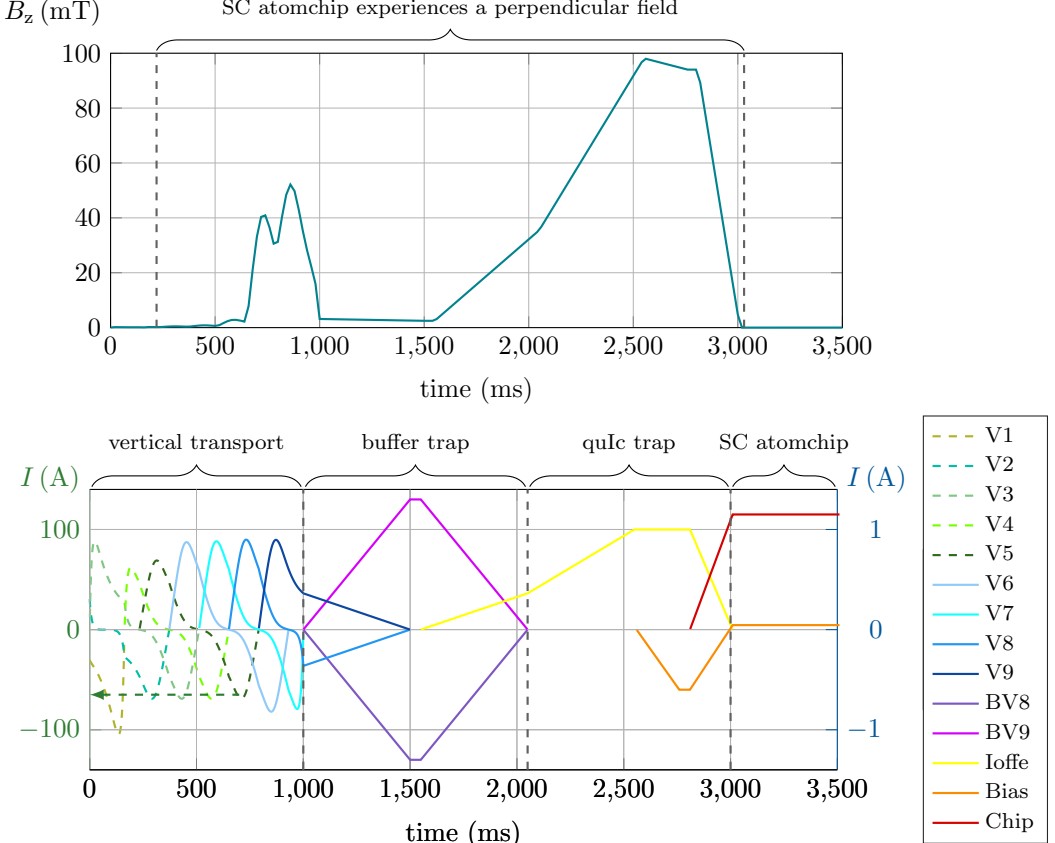

Figure 3: Magnetic field history (Top), perpendicular to the film experienced by the atomchip at the middle of the Z structure. The fields are calculated from the current sequence of the experiment (Bottom). Only the last sequence of the magnetic transport is shown which is the vertical section of the transport. The vertical section is split into the normal- (dashed lines) and super-conducting (solid lines) coils. The normal conducting coils go up to 100 A (left axis) whereas the superconducting coils/wire only go up to 2 A (right axis). This is proceeded by loading into a buffer quadrupole trap and subsequently into a quadrupole-Ioffe trap (quIc trap). The final seqence is the loading into the superconducting atomchip where a bias field and the transport current for the niobium wire is ramped up to a desired value that creates the trap. Details of the magnetic transport and loading schemes are discussed in detail in [37]. The niobium film experience a maximum field of almost 100 mT.

In the following subsections, we show that the current-induced magnetization has a significant effect in our niobium wire used for our magnetic trapping experiments.

To understand the current distribution through the wire cross-section of the Z structure, we apply the Brandt model [40]. The model is widely used for describing the current distribution for type-II superconducting thin-films for an applied current and/or magnetic field. It is mainly based on solving the London equations and applying the CSM. As mentioned earlier, the central part of the 2 mm long Z (figure 2 inset) can be treated as a thin rectangular cross-section extending to infinity. Figure 4 shows the results of the Brandt model for a transport current on a rectangular thin superconducting film extending to infinity. The film starts from the virgin state where no magnetization is present. The transport current $I_t/I_c$ is then increased from 0 to 0.88 (figure 4 left) and down to $-0.88$ (figure 4 right) where $I_c$ is the critical current. At $I_t/I_c = 0$ after having reached a maximum current of $I_t/I_c = 0.88$, there is a non-zero current

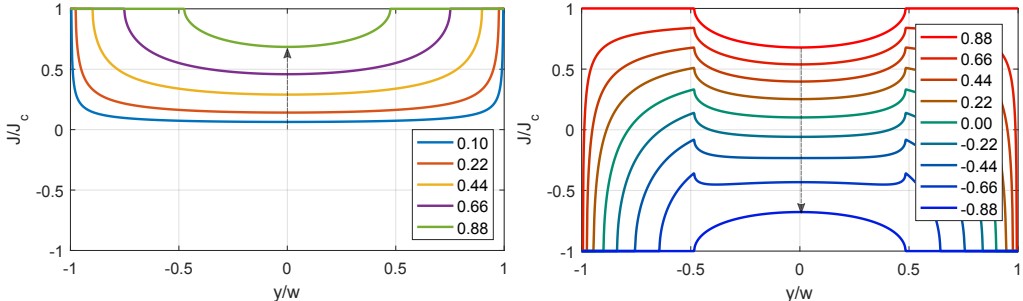

Figure 4: Current distribution profile forming in a rectangular cross-section super-conductor extending to infinity: (Left) shows the current distribution across the entire wire width $2w$ increased from the virgin state to a maximum transport current of $I_t/I_c = 0.88$ and (Right) when decreased down to $I_t/I_c = -0.88$. The arrows indicate the direction of current change.

distribution across the wire width. This is the current-induced remanent magnetization of the superconducting film. In this simple model, the cross-sectional current distribution of the wire depends entirely on the previously achieved maximum applied current.

## 4.2 Comparison to experiment

We will now apply the Brandt model for a purely current-induced magnetization to our full set of experimental data for various applied transport currents from $1.9\,\text{A}$ down to $0\,\text{A}$ using the niobium atomchip shown in figure 1. As illustrated in figure 5, the Brandt model for $I_{max} = I_c$ describes the full set of measurements for all transport currents for $I_c = 8\,\text{A}$. The inset of figure 5 shows the current distributions used to describe the experimental data obtained from the Brandt model. The fitted $I_c = 8\,\text{A}$, is higher than the $I_c$ measured for our niobium structure ($I_c = 3.5\,\text{A}$). We conjecture that the difference comes from the fact that the Brandt model and CSM gives only an approximate description of the remanent magnetization of a niobium film [48–50].

In our measurements, we always ramp to the target current from zero. There is no prior application of a higher current to create a different history. The current-induced remanent magnetization is established before the experiment begins. We apply a current exceeding $I_c$ to measure it. This also establishes the one single current maximum $I_{max} = I_c$ for a whole set of experiments.

Ideally, exceeding $I_c$ should quench the film. However, in a set of separate experiments, it was found that the remanent magnetization remained in the central section of the Z-wire even after a quench by exceeding $I_c$. We suspect that we are only quenching the weak spots of our niobium film which are the regions with the Aluminum bonds (See figure 1b). In order to fully erase the remanent magnetization, the cryostat has to be warmed up significantly. We confirmed the quench by reproducing the field-induced remnant magnetization trap in [26] or our current-induced remanent magnetization trap until no trap was formed. The specific details are discussed thoroughly in [38, 51–53].

Further evidence on the relevance of the current-induced magnetization is by looking at the position of the trap relative of the wire center. This is illustrated in figure 6 by absorption imaging of the atoms through the longitudinal direction (x-axis in inset of figure 1).

Current- and field-induced magnetization show a different symmetry of the current distribution across the wire cross-section [23, 40]. Field-induced magnetization is asymmetric whereas current-induced is symmetric. This means the trap formed by a current-induced magnetization moves symmetrically straight towards the center of the wire-width as the bias field

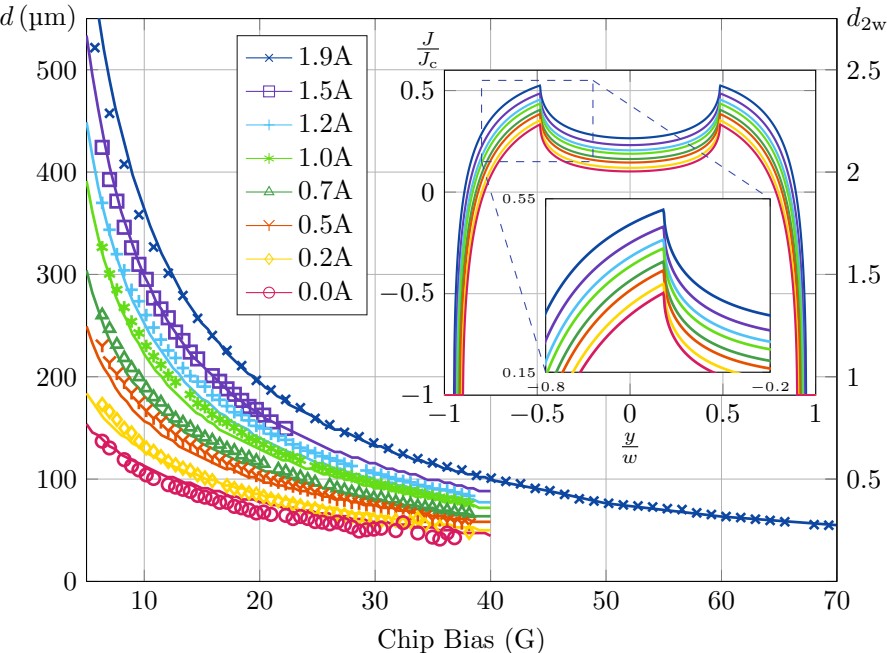

Figure 5: Probing the distance of the atomcloud to the chip surface at different transport currents for the same previously reached current maxima: The discrete markers are experimentally measured distances versus bias field for various currents down to zero transport current. The presence of the zero-current-trap indicates the presence of remanent magnetization in the superconducting film. The solid curves are fits using the current history assuming a previously reached $I_{max}/I_c = 1$. The current distribution used for the fits is shown in the inset for the respective transport current. See text for more details.

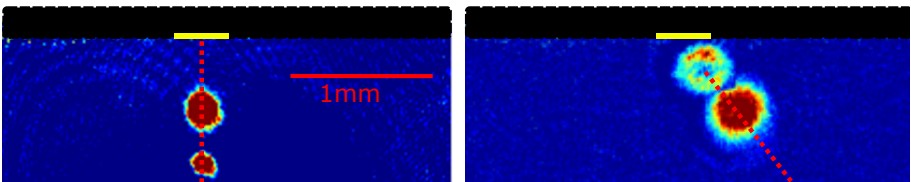

Figure 6: Absorption composite images in the longitudinal direction to illustrate the difference between current- and field- magnetization: The trajectory of the atoms towards the trapping wire for increasing bias field are illustrated for the case of a purely (Left) current-induced magnetization and (Right) field-induced remanent magnetization. The latter is a pure remanent trap without any transport current similar to [24, 26] where $H_{c1}$ is intentionally exceeded. The red dashed line illustrates the direction of the atoms for increasing bias field which leads to the superconducting trapping wire. The atomchip surface is indicated by the black region where the estimated location of niobium wire cross-section is illustrated in yellow. The composite images are for a bias field of 10 G (1.00 mT) and 30 G (3.00 mT), respectively.

increases (See figure 6 left). The trap simply moves normal along the center line of the film just as in any conventional atomchip experiment. Field-induced remanent magnetization traps have an asymmetric current distribution, which means that the trap moves along an angle with respect to the normal vector at the center of the film wire width [22, 24]. This is illustrated in figure 6 right for a pure field-induced remanent magnetization trap. Any significant additional

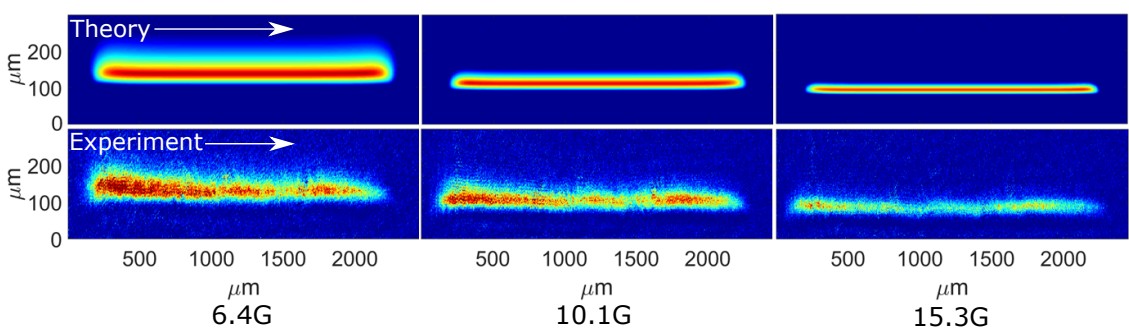

Figure 7: Zero-current-trap from left to right for a bias field of 6.4 G (0.64 mT), 8.9 G (0.89 mT), and 12.7 G (1.27 mT). (Top), simulated reverse potentials shown in colormap where blue to red corresponds to 5 G (0.50 mT) and 0.5 G (0.05 mT), respectively. The simulated potentials are calculated for the full 2D Z-wire structure using the current distribution in the inset of figure 5 inset for zero current all through out the wire cross-section. (Bottom), absorption image of the atomic cloud with a temperature 42 μK.

field-induced magnetization along with the current-induced magnetization will manifest itself by tilting of the atomcloud trajectories.

The magnetic transport in our setup produces magnetic fields (figure 3) which could induce a remanent magnetization. Superconducting films with thickness in the order of the London penetraion depth can lead to an induced ramanent magnetization even with an applied field much lower than $H_{c1}$ as suggested by [54]. In our case, the film is thicker than the London penetration depth, we infer that the demagnetization effect could lead to field-induced magnetization. Our niobium film structure has a demagnetization factor of 0.9975 [55], which means the field will be amplified by a factor 400 at the film edges. Despite this, the atom-cloud trajectories in the experimental situations as shown for figure 5 and 2 are moving purely normal to the film, which means that the current distribution across the wire cross-section is mainly symmetric and defined by the current history. This observation that the remanent magnetization in a niobium wire is mainly current-induced can be at least qualitatively understood.

Finally we now discuss the zero current trap formed by the current history. Figure 7 shows a comparison between the atomcloud images and the trapping potential simulations for the current-induced zero-current trap for different applied bias fields. This trap is a remanent magnetization trap similar to [21–24,26], only that the magnetization is induced by a transport current. The measurement matches the simulation, shown in an inverted potential colormap, quite well except for a small tilt in the trap bottom of the cloud. This tilt is not observed in traps with transport current. A more detailed numerical study for a superconducting Z-wire including the zero-current trap was conducted by Sokolovsky et al. [56] and also does not show this asymmetry. Their results for the zero-current trap on a very wide Z-wire with a central length of only $3 \times 2w$ suggest a strong double trap where two minima split towards the corners of the Z-wire. Our zero-current trap only shows a hint of this effect at large bias fields (trap close to the chip surface). This much smaller double zero current trap is caused by the longer central length ($10 \times 2w$) of our Z structure.

# 5 Conclusion and Outlook

In this work, we present evidence of a current-induced remanent magnetization in the niobium structure through ultracold $^{87}$Rb atoms. Our experiment shows that current-induced magnetization is the dominant source of magnetization and is important to consider when performing atom trapping with niobium structures. Due to to this effect, our wide $200\,\mu$m wire's current flow resembles that of a thin wire. The CSM based Brandt model by [40, 46] is found to be sufficient in describing a single current maximum history for the $500\,$nm thick niobium film. This opens up the possibility of creating unique remanent magnetization traps by controlling the transport current history of the superconducting film [6, 35, 57]. It will also pave the way to new and novel magnetic traps for ultracold atoms utilizing superconducting properties.

## Acknowledgements

The authors would like to acknowledge the insightful discussions with Michael Trupke, Thorsten Schumm, Alvar Sanchez and Oriol Romero-Isart. The authors would also extra appreciation for the superconductivity discussions with Franz Sauerzopf. The authors would also like thank Thomas Schweigler, Bernhard Rauer, Mira Maiwöger, Mohammadamin Tajik and Joao Sabino for helping proof read the manuscript.

**Author contributions** FD and SM performed the experiment with the help of TW, BG, NS and ŻK. FD wrote the initial manuscript with all authors contributing in the final stage. JS conceived the experiment.

**Funding information** The authors would like to acknowledge the support of the CoQuS doctoral program. This work was funded through the European Union Integrated Project SIQS and the Austrian Science Fund FWF (Wittgenstein Prize, SFB FOQUS).

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
