# Peer review of "Current-induced magnetization hysteresis defines atom trapping in a superconducting atomchip"

_SciPost Physics, doi:SciPost Phys. 4, 036 (2018)_

## Round 1 · Referee Report · Anonymous · 2018-4-12

Strengths

see in the report

Weaknesses

see in the report

Report

Diorico et al. describe the behaviour of magnetic traps for neutral formed by external bias fields and currents in a superconducting Niobium wire. They report that the distance of the trap minimum as a function of the bias field does not follow the behaviour of an expected Meissner current sheet. The article identifies transport current-induced magnetization that leads to currents flowing in the center of the wire as the main reason for the observations. The authors demonstrate good agreement between their data and Brandt's extension of the Bean model.

While the penetration of flux into superconducting films as a result of transport currents is nothing new or unexpected, the authors are the first to systematically study it in the context of cold atom traps. I am confident that the results are interesting for the community working on superconducting atom chips and support the publication of the manuscript.

The following comments may help the authors to fine tune the manuscript:

1. It seems to me that the insert mentioned in this footnote "see also insert of figure 1" is missing.

2. Sect. 3, Figure 2 and Figure 5:
I find it difficult to understand the data because the history of the current driven through the Z-wire does not become clear. I recommend extending on this issue.

Which current was driven through the wire to load the atoms?
As far as I understand - I might be wrong, because it is not clearly stated -, the sequence is as follows:
- At the end of transport, the atoms are loaded into a chip trap by driving a current I_load = XXX and a bias field B_load.
- The current is then reduced to I_t and the field to B_bias, and the position of the atoms is measured.
- The sequence is then repeated.

Is the chip heated up above T_c to start the experiment with a film free of flux? If not, multiple sequences of high and low currents are driven through the chip, making the current distribution much more complicated than indicated in Figure 4.

3. Figure 3:
- This graph is hardly traceable without an explanation of the color coding.
- Label: Magnetic transport Is this the driven current in A?
- The bottom graph shows perpendicular fields on the order of 100mT. Is this really below Hc1 at 4K?

4. Section 4.2:
How does the fitted current of I_C = 8A come about? Without any explanation of the fitting procedure and understanding of the current history, this remains unclear.

How is the trap loaded in case of the pure field-induced magnetization trap?

Last paragraph of 4.2: "The measurement matches the simulation [...] except for a small tilt in the trap bottom of the cloud. This tilt is not observed in traps with transport current." It is not clear to me what the authors would like to express in theses sentences and the rest of the paragraph.

5. References:
The authors may review and unify the formatting of the references.

6. Language:
Please check wether the expression "remnant magnetization" is adequate or "remanent magnetization" would be better.

7. A couple of typos have caught the eye:
p.5 section 4: type II --> type-II
p.5 section 4: remnanat --> remnant or remanent
p.6 section 4: history the magnetic fields --> history of the magnetic fields
p.6. Figure 3: niobioum --> niobium
p.6 sect. 4.1: based on the solving --> based on solving

Requested changes

see in the report

  • validity: high
  • significance: good
  • originality: good
  • clarity: good
  • formatting: good
  • grammar: good

Author:  Fritz Diorico  on 2018-05-26  [id 259]

(in reply to Report 1 on 2018-04-12)
Category:
answer to question
correction

We would like to thank the referee for his/her valuable feedback and insight to our paper. Below we address answer the questions and errors from the report.

  1. We intended to point out the inset in figure 2 showing definition of the wire width as 2w. We have made the correction.

  2. We missed out on this detail. We added a sentence in figure 2 and two paragraphs in section 4.2 to clarify this. We always load the trap starting from zero current. The atoms are loaded directly from ramping into the target transport current, $I_{t}$, without any intermediate loading current.

In our experimental procedure a high transport current was always applied before the experiment in order to check the film's critical current. This determined and set the imprinted current-induced magnetization which determines the rest of the measurements where $I_{max} \approx I_{critical}$. The calculated curves are based on this experimental procedure: the maximum current was already achieved beforehand. To erase or change the imprinted remnant magnetization, we would have to exceed this previously applied maximum current $I_{max}$.

This imprinted magnetization could be reset by exceeding the critical current, which should quench the entire superconductor. It was found that exceeding the critical current did not remove the remnant magnetization of the niobium film. The most likely explanation for this is that the quench comes from the heat produced by the Aluminum bonds (not superconducting at our operating temperature of about 5K). In our case, the Z-wire on the chip used for trapping is very well thermally connected to the sapphire substrate which itself is connected to a massive heat sink at the 4K stage of the cryostat, and the central region of the Z-wire never quenches. In order to fully erase the remanent magnetization, the cryostat has to be warmed up significantly. We confirmed this by reproducing the field-induced remnant magnetization trap like the ones explored by R. Dumke’s group in [26].

  1. We apologize for not clearly distinguishing the curves. The main message of Fig. 3 is the perpendicular fields experience by the niobium film by the sequence of fields used to prepare the ultracold sample on the atomchip (as described in our publication about our experimental setup ref [37]).

The magnetic transport is composed of two sections: a section outside the cryostat built from normal conducting coils and one inside the cryostat built with superconducting coils. There is a massive difference in current for the normal-and super-conducting sections due to current limitations in the superconducting section. The normal conducting section has currents that go up to 200A whereas in the superconducting section only go up to 2A. However, they all maintain the same gradient since the superconducting coils are compensated with more windings (~3000 each transport coil) whereas for the each normal conducting transport coil has only 30 windings. We added a second axis (left) for the current in the normal conducting coils and (right) for the current in the superconducting coils.

The magnetic field plot is Biot-Savart calculation for the entire magnetic coil system with all the magnetic transport sequences, mainly intended to infer how much perpendicular fields that the niobium film experiences during an experimental atom loading sequence. We also corrected a small calculation error in the perpendicular fields. The sequence is now plotted with more clarity. We also adapted the figure caption to provide a better explanation.

  1. In connection the reply in question 2, the previously reached maximum current ($I_{max}$) experienced by the niobium film determines the remanent magnetization. Any subsequent applied current ($I_{a}$) that is lower that Imax, will not change the remanent magnetization and thus determine all the current distributions for $I_a<I_{max}$. In order to fit the data, we assumed that $I_{max}=I_{c}$. As discussed in the reply of question 2, this matches our experimental cycle. The only part we could not understand is that $I_{c}=I_{max}=8A$. Any lower, the data doesn’t fit. With this assumed $I_{max}$, all data for any transport current lower than $I_{max}$ fit very nicely. Our film with its cross-section has a measured $I_{c}$ of 3.5A at around 4.2K, which is a factor 2 smaller then estimated from the remnant current model. We think that this difference could be caused by the limitation of the Brandt vortex model which is beyond the scope of this study. Despite the limitation, it still allows us predict the behavior of the atom traps created by our niobium wire.

To load into the pure field-induced magnetization trap, the loading sequence is more or less similar to the regular chip loading sequence. We bring the atoms close to the trapping wire from the quadrupole-Ioffe trap and slowly ramp it down while ramping up the appropriate bias field to create the remanent magnetization trap. It is very similar to the remanent magnetization traps demonstrated by R. Dumke in Singapore with his YBCO superconducting atomchips [26].

In our absorption images with the zero-current-remanent trap, we observed an increased atomic density favoring one side of the trap. We only observed this with the zero-current-remanent trap but not with the traps formed by a transport current. We do not know the underlying mechanism for the slight trap bottom tilt. It is also not observed in a more detailed numerical study done by 3D modeling of magnetic atom traps on type-II superconductor chips [56]. We reformulated this paragraph to make it clearer.

  1. – 7. We have made the appropriate corrections and also adapted “remanent” as also suggested by the second referee.

---

## Round 1 · Referee Report · Anonymous · 2018-4-30

Strengths

Please refer to report

Weaknesses

Please refer to report

Report

This manuscript describes the experimental investigation of remanent magnetic fields in superconductors, using ultracold atoms. When atoms are trapped by a current-carrying wire, the trap position over applied bias field is a function of the current distribution in the wire. Here, the authors are using this fact to probe the current distribution in a superconducting Niobium wire. They find that in their case the distribution is not given by an expected Meissner-London surface current, but by a narrower distribution. Using Brandt’s vortex model they show that the modified current distribution can be explained by transport current induced magnetization of the trapping wire.
Even though these results are expected and well described in the superconductor community, I support the publication of this manuscript as it describes a novel phenomenon in atom chip technologies. Overall the manuscript is written well however it may require a few minor changes.

Requested changes

• It might contribute to the readability of the manuscript to separate Figure 1 in two parts, for instance (a) and (b). In the text the Nb chip is usually referred to as the ‘inset of Figure 1’, but I also found that it was referred to as ‘figure 1’.
• On page 5 the author claim ‘Unlike the Meissner current trap discussed above, the trap remains closed for increasing applied bias field up to the fundamental limit (Casimir-Polder)’. Is there evidence to support this claim ? I would expect that at the observed distances, the Casimir-Polder interaction is negligible. At the same time the remanent magnetization has a certain width, so the current distribution might still be the limiting factor.
• It is written that during the experimental sequence the lower critical field is never reached. Later, a vortex trap generated by vortices from an external field is shown. Is it possible to deduce the lower critical field from this measurement ? Meaning the magnetic field from which on a vortex trap can be generated ? If this would be known it would be possible to estimate the critical transport current at which vortices start to penetrate the wire. As the wire in this publication has a thickness on the order of 10 times the London length, a correction factor might be necessary to calculate the critical current. Details for this can be found for instance in this publication (and references therein) : Universal self-field critical current for thin-film superconductors. Nat. Commun. 6:7820 doi: 10.1038/ncomms8820 (2015).
• Figure 5 : the inset graph is very unclear as the curves are overlapping. It might be helpful to reduce the number of curves, or to offset them.
• Figure 7 : it is mentioned that the potential is calculated from the current distributions shown in the inset of figure 2. This should be figure 5 ?
• Concerning the comparison to the Brandt model. I couldn’t find a detailed description of the experimental procedure. Especially not the history of the current ramps. From the simulations I guessed that the current was increased to the critical current and then decreased to a certain final current. Looking at figure 3 though it seems that, while loading the chip trap, the current actually starts from 0. In this case the sequence would be : increasing to the critical current -> decreasing to 0 -> increasing to final current. It should be made clear which procedure was used, as the simulations don’t apply to the second sequence.

• In general I believe ‘remnant’ should be ‘remanent’, on page 5 second to last paragraph I also found ‘remnanat’, which seems to be a hybrid word.
• Page 5, paragraph 4 : ‘For atomchip traps created by an applied transport currents …’
• Page 9, paragraph 1 : ‘To create field-induced remnant traps Hc1 needs to exceeded…’
• Figure 6 caption : Hc should be replaced by Hc1 ?

  • validity: top
  • significance: good
  • originality: good
  • clarity: high
  • formatting: excellent
  • grammar: good

Author:  Fritz Diorico  on 2018-05-26  [id 260]

(in reply to Report 2 on 2018-04-30)
Category:
answer to question
correction

We thank the referee for his/her insightful report on our paper. Below we address the all points in the report. We hope to have address the important points.

We have separated figure 1 into two parts for clarity. For figure 5, we have added small zoomed-in view of the curves. The curves look very similar and the colors correspond directly to the experimental data and fitted curves. We made the correction for figure 7. It should be referring to the inset of figure 5. We have corrected the typographical errors and have adapted the term "remanent" which was also suggested by referee 1. - For the fundamental limit query: Yes, for trapping ultracold atoms on an atomchip the Casimir-Polder force is the ultimate limit. Apart from technical noise in conductors, the main limiting factor prohibiting magnetic traps from coming close to the atomchips is the Casimir-Polder interaction. If the atoms get very close (typical below 1 micrometer) then the magnetic trap cannot compensate the strong attraction from the Casimir-Polder force. This has been investigated in detail by the Vuletic group. We added Lin et al., PRL 92, 050404 (2004) as ref [42]. We did not measure traps down to the sub-micometer distance, so we did not directly check that the trap is closed until the final Casimir-Polder limIt. Never the less the scaling of the height with bias field, which follows a thin wire, is a good indicator that it remains closed and tightly confining up to close distance to the surface. A different indicator for the trap remaining closed is the fact that the current flows in center. The Meissner current trap opens up because the current flows on the edges. We took the inferred current distribution from fig 4 and calculated the trap down to sub-micron distances, and confirmed this conjecture. We thank the referee for pointing out that this was not clearly stated in the old manuscript, and we added the appropriate references to our original statement and a sentence in the conclusion. - For the field-induction at lower fields query: Thank you very much for pointing us to this reference as ref [54] and found in addition [55]. Since our film thickness is much larger than the London penetration depth, we infer that, even if we do not exceed Hc1, we might still have induced some field-type magnetization mainly from demagnetization effects. Our film is 500nm thick and 200microns wide. Following references ref. [54,55], this gives us a demagnetization factor of 400 for our thin-film, [55] (Demagnetizing factors of rectangular prisms and ellipsoids, Du-Xing Chen, E. Pardo and A. Sanchez, IEEE Tran. Magnetics, Vol 38, No 4). At the edges of the film, the applied field should be magnified by this amount. Hence, there might always be a remanent magnetization from magnetic fields if the applied field exceeds Hc1/400. This will add a tilt in the motion of the trap towards the chip (see discussion in section 4 and fig 6). As shown in Fig 6 our measurements indicate that the current-induced magnetization is more significant.
We added a short test discussing this and include now the suggested citation in section 4.2 - For the query on comparison to the Brandt model and the history experienced by our niobium film: We thank the referee to pointing us to a passage that was indeed not very clear. Yes, we missed to clearly state these details. We added now a sentence in figure 2 and two paragraphs in section 4.2 to clarify this. We always load the trap starting from zero current. But in our experimental procedure a high transport current was always applied before the experiment in order to check the film's critical current. This determined the imprinted current-induced magnetization which determines the rest of the measurements where $I_{max} \approx I_{critical}$. All the calculated curves are based to this experimental procedure: the maximum current was already achieved beforehand. To erase or change the imprinted remnant magnetization, we would have to exceed this previously applied maximum current $I_{max}$. This imprinted magnetization could be reset by exceeding the critical current, which should quench the entire superconductor. It was found that exceeding the critical current did not remove the remnant magnetization of the niobium film. The most likely explanation for this is that the quench comes from the heat produced by the Aluminum bonds (not superconducting at our operating temperature of about 5K In our case, the Z wire on the chip used for trapping is very well thermally connected to the sapphire substrate which itself is connected to a massive heat sink at the 4K stage of the cryostat, and the central region of the Z wire never quenches In order to fully erase the remanent magnetization, by quenching the entire film the cryostat has to be warmed up significantly We confirmed this by reproducing the field-induced remnant magnetization trap like the ones explored by R. Dumke’s group in [26].

---

## Round 3 · List of Changes

Summary of changes: 1) We have modified figure 1 for better clarify as per referee suggestion. 2) Figure 3 and its caption have been reformatted in order to better understand the experimental sequences and as well as the magnetic field history of the atomchip. The magnetic field history was also recalculated since there was an error in the original plot.
3) We have made changes to section 4.2 to include the referee queries. We believe these will be useful to the readers. We have added a few paragraphs after the first original paragraph to explain the history of the experienced currents and fields of the niobium atomchip which was not explained in the original submission. In connection to this, a short sentenced is also added in figure 2 mentioning this briefly. We rewrote paragraph 4 on the original submission to include the suggested reference by referee 2. We have also rewritten the last paragraph to better explain the zero current trap. 4) We added reference [42] in paragraph 3 of section 3 which is in connection to the referee 2’s query. 5) We added a sentence in the conclusion and outlook section to highlight the overall effect of the current-induced magnetization to how the wire behaves. 6) We have made the typographical errors pointed out by the referees throughout the text and have adapted the term “remanence” as suggested

---

## Editorial Decision

published